# Impaired speed encoding and grid cell periodicity in a mouse model of tauopathy

Thomas Ridler[1]*, Jonathan Witton[1,2], Keith G Phillips[3], Andrew D Randall[1,2], Jonathan T Brown[1]*

[1]Institute of Biomedical and Clinical Sciences, University of Exeter Medical School, University of Exeter, Hatherly Laboratories, Exeter, United Kingdom; [2]School of Physiology, Pharmacology and Neuroscience, University of Bristol, University Walk, Bristol, United Kingdom; [3]Lilly United Kingdom Erl Wood Manor Windlesham, Surrey, United Kingdom

**Abstract** Dementia is associated with severe spatial memory deficits which arise from dysfunction in hippocampal and parahippocampal circuits. For spatially sensitive neurons, such as grid cells, to faithfully represent the environment these circuits require precise encoding of direction and velocity information. Here, we have probed the firing rate coding properties of neurons in medial entorhinal cortex (MEC) in a mouse model of tauopathy. We find that grid cell firing patterns are largely absent in rTg4510 mice, while head-direction tuning remains largely intact. Conversely, neural representation of running speed information was significantly disturbed, with smaller proportions of MEC cells having firing rates correlated with locomotion in rTg4510 mice. Additionally, the power of local field potential oscillations in the theta and gamma frequency bands, which in wild-type mice are tightly linked to running speed, was invariant in rTg4510 mice during locomotion. These deficits in locomotor speed encoding likely severely impact path integration systems in dementia.

*For correspondence:
t.ridler@exeter.ac.uk (TR);
j.t.brown@exeter.ac.uk (JTB)

## Introduction

Accurate spatial navigation requires the integration of sensory information to generate neural representations of space. Various high-level representations of the external environment are expressed at a single-cell level within the extended hippocampal formation (hippocampus proper, dentate gyrus, entorhinal cortex and subiculum) and connected brain areas, regions well known to be critical for spatial memory. For example, place cells in the hippocampus fire action potentials in specific spatial locations (*O'Keefe and Dostrovsky, 1971*), whereas grid cells in the medial entorhinal cortex (MEC) fire in a highly organised, hexagonally distributed spatial pattern across an environment (*Fyhn et al., 2004*; *Hafting et al., 2005*). These directly spatially sensitive neurons are collocated within the MEC with other functionally defined cell types, including head-direction (HD; *Giocomo et al., 2014*; *Sargolini et al., 2006*; *Ranck, 1984*; *Valerio and Taube, 2012*), speed (*Kropff et al., 2015*; *Hinman et al., 2016*), and border cells (*Lever et al., 2009*; *Solstad et al., 2008*). Together these are thought to provide the crucial computational information required for effective path integration, the process of using idiothetic cues to continuously calculate positional information (*Barry and Bush, 2012*).

Patients with Alzheimer's disease (AD) and other dementia-spectrum disorders exhibit profound disruption in spatial navigation and memory, even at very early stages of the disease (*Allison et al., 2016*; *Lithfous et al., 2013*; *Hort et al., 2007*; *Laczó et al., 2011*; *Mokrisova et al., 2016*). At a pathological level, misfolded tau deposition typically occurs first in the entorhinal cortex and

hippocampus (*Pooler et al., 2013*). Taken together, these clinical signs strongly implicate pathology-induced circuit-level dysfunction in the hippocampal formation as a key early-stage functional deficit in AD. In this regard, there is substantial evidence from transgenic mouse models that dementia pathologies, such as β-amyloid deposition (Aβ) and hyperphosphorylation and misfolding of tau, can disrupt the intrinsic properties (*Booth et al., 2016a*; *Tamagnini et al., 2017*; *Brown et al., 2011*; *Kerrigan et al., 2014*) and synaptic microcircuits (*Hoover et al., 2010*; *Brown et al., 2005*; *Fitzjohn et al., 2001*) of pyramidal cells in area CA1 of the hippocampus. Furthermore, there is growing evidence that place cells in the CA1 region of APP and tau overexpressing mice have reduced spatial-sensitivity (*Booth et al., 2016a*; *Cheng and Ji, 2013*; *Cacucci et al., 2008*), strongly suggesting a failure of some aspects of the upstream functional circuits involved in spatial cognition. Recent evidence suggests that the circuits required for generation of theta and gamma frequency oscillations in the dorsal entorhinal cortex are especially prone to dysfunction in a mouse model of tauopathy (rTg4510 mice) (*Booth et al., 2016b*). Furthermore, *Fu et al., 2017* demonstrated that grid cell spatial periodicity is reduced in mice where tau overexpression is restricted to the entorhinal cortex. These data correlate with human imaging studies which suggest deficits in grid-cell-like activity in the entorhinal cortices of people at genetic risk of developing AD (*Kunz et al., 2015*).

Importantly, the effect of tau pathology on other functional components of the spatial navigation system is less well understood. In this study, we report for the first time that disruption to speed encoding in the MEC of rTg4510 mice may underlie deficits in grid cell function. Using high-density silicone probes and tetrode recording approaches in freely moving animals, we report that in both the MEC and the CA1 region of the hippocampus, encoding of speed information at both the local field potential and cellular level is substantially impaired.

## Results

### Tau pathology is associated with loss of oscillatory speed coding

Neural coding of spatial information is likely to require the precise representation of locomotor speed (*Barry and Bush, 2012*; *McNaughton et al., 2006*; *Burak and Fiete, 2009*; *Bush and Burgess, 2014*). Velocity information can be represented in the brain via the dynamic regulation of the power and frequency of neuronal network oscillations in the theta and gamma frequency bands (*Ahmed and Mehta, 2012*; *McFarland et al., 1975*; *Zheng et al., 2015*; *Chen et al., 2011*; *Sławińska and Kasicki, 1998*). We hypothesised that the profound deficits in spatial memory that occur in response to tau pathologies (*Booth et al., 2016a*; *Fu et al., 2017*; *Ramsden et al., 2005*) arise from impaired representation of locomotor speed in the hippocampal formation, ultimately leading to deficits in the encoding of spatial information. To examine this hypothesis, we implanted multi-site silicone-based recording probes in the dorsal MEC of male wildtype (WT) and rTg4510 mice (6–7 months). Following a post-surgical recovery period, mice were encouraged to explore a familiar linear track, baited by food rewards at either end to encourage running between the two ends, whilst connected to a multi-channel electrophysiological recording apparatus via a lightweight tether cable.

As expected, in WT mice, theta oscillation properties followed changes to locomotor activity (*Figure 1A/B*). Pooled data illustrate a clear relationship between running speed and theta oscillation power (linear regression: $R^2$ = 0.71, p<0.001, n = 7 mice) and frequency (linear regression: $R^2$ = 0.69, p<0.001, n = 7 mice) (*Figure 1Ci*). In contrast, in rTg4510 mice, theta oscillation amplitude was poorly correlated with locomotor activity and remained at consistent levels throughout recording sessions (linear regression: $R^2$ = 0.15, p=0.02, n = 8 mice, *Figure 1Ci*). Importantly the overall Z-transformed correlation coefficient and slope relationships for theta power-running speed relationships were significantly lower in rTg4510 mice when compared to WT controls (two-way repeated measures ANOVA, see *Table 1* and *Figure 1D*). Although less pronounced than in WT animals, peak theta frequency was correlated with running speed in rTg4510 mice (linear regression: $R^2$ = 0.74, p<0.01, n = 8 mice, *Figure 1Cii*), which across the population was not significantly different from WT (Correlation (Z'); WT: 0.84 ± 0.24, rTg4510: 0.40 ± 0.14, unpaired T-test, p=0.13, n = 7/8 mice), with no difference in the slope of the frequency-running speed relationship (Correlation (Hz/cms$^{-1}$); WT: 0.045 ± 0.0077, rTg4510: 0.037 ± 0.011, unpaired T-test, p=0.53, n = 7/8 mice).

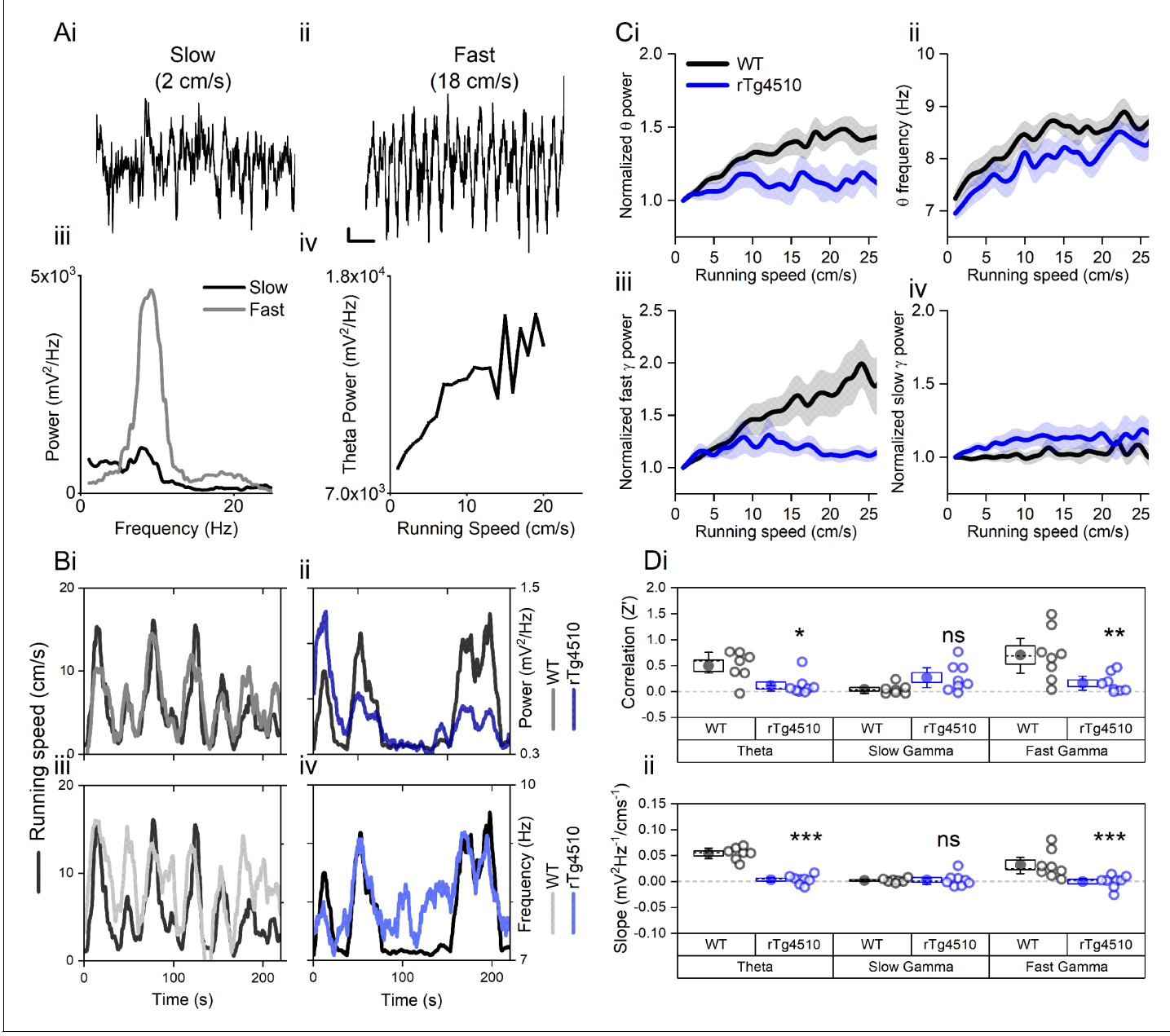

**Figure 1.** Oscillation-running speed relationship is impaired in rTg4510 mice. (A) Local field potential from WT mice showing periods of slow (i) and fast (ii) running speed showing faster and larger theta oscillations during locomotor activity. (iii) Power spectra of data shown in i and ii for slow (black) and fast (grey) running periods. (iv) Example relationship between running speed and average theta oscillation power across recording session. (B) Example plots showing animals running speed on linear track (black, left Y axis) showing high correlation with theta oscillation (grey, right Y axis) power (i) and frequency (iii) over several minutes of recording. Corresponding example from rTg4510 mouse showing with theta oscillation amplitude (ii) and frequency (iv) with decreased association with running speed. (C) Running speed-theta oscillation relationships for power (i; normalised to 1–2 cm/s bin) and frequency (ii). Also shown are fast (iii) and slow (iv) gamma power-running speed relationships. (D) Pooled data for each animal showing Z-transformed correlation coefficients (i) and slopes (ii) of running speed-oscillatory power relationships for different frequency bands (*p<0.05, **p<0.01, ***p<0.001, ns = not significant, Bonferroni-corrected pairwise-multiple comparisons; for 2-way repeated measures ANOVA main effects and interactions see *Table 1*).

The online version of this article includes the following figure supplement(s) for figure 1:

**Figure supplement 1.** rTg4510 mice display hyperactive phenotype.

**Table 1.** Results of two-way repeated measure ANOVA for oscillation-running speed interaction. Results of relevant post-hoc pairwise-comparisons displayed on *Figure 1*.

**Correlation (Z)**

| Source of variation | Df | Sum-of-squares | Mean square | F | p |
|---|---|---|---|---|---|
| Frequency | 2 | 0.470 | 0.235 | 4.341 | 0.051 |
| Genotype | 1 | 0.602 | 0.602 | 5.927 | 0.038 |
| Interaction | 2 | 1.330 | 0.665 | 8.379 | 0.005 |

**Slope (mV2/Hz.cm$^{-1}$ )**

| Source of variation | Df | Sum-of-squares | Mean square | F | p |
|---|---|---|---|---|---|
| Frequency | 2 | 0.004 | 0.002 | 8.973 | 0.004 |
| Genotype | 1 | 0.008 | 0.008 | 86.224 | >0.001 |
| Interaction | 2 | 0.005 | 0.003 | 16.623 | >0.001 |

Similar running speed modulation has been observed in both in the fast (60–120 Hz) and slow (30–50 Hz) gamma frequency bands (*Zheng et al., 2015*; *Chen et al., 2011*). In WT mice, both fast and slow gamma oscillation power was positively correlated with running speed (linear regression; fast gamma: $R^2$ = 0.90, p<0.001, n = 7 mice; slow gamma: $R^2$ = 0.17 p=0.013, n = 8 mice), although the slope of this association was greater for fast gamma frequencies (slow gamma: 27.1 ± 8.4 mV$^2$/Hz.cm$^{-1}$, fast gamma: 80.6 ± 19.2 mV$^2$/Hz.cm$^{-1}$; two-way repeated measures ANOVA; *Table 1* and *Figure 1D*). rTg4510 mice did not show significant correlations for fast gamma frequency band oscillations (linear regression; fast gamma: $R^2$ = 0.03, p=0.67, n = 8 mice, *Figure 1D*), but did for slow gamma frequencies ($R^2$ = 0.43, p<0.01, n = 8 mice, *Figure 1D*). However, compared to the WT population, this correlation was significantly lower only in the higher gamma frequency range (2-way repeated measures ANOVA, *Table 1* and *Figure 1D*, n = 7/8 mice).

rTg4510 mice display hyperactive phenotype rTg4510 mice have been shown to display a hyperactive phenotype under various conditions, which becomes more pronounced with developing tau pathology (*Selenica et al., 2014*; *Cook et al., 2014*; *Jul et al., 2015*). Since the experiments shown here display data that is heavily influenced by running speed, it was therefore important to observe this effect in the current experimental subjects. Under these recording conditions, rTg4510 mice also displayed a hyperactive phenotype. rTg4510 mice were shown to spend more time at faster running speed, with average speeds across recording sessions greater than WT control mice (p=0.04, Unpaired T-test, n = 8/10, *Figure 1—figure supplement 1*).

## Firing properties of MEC single units

Impairments to the theta oscillatory code for running speed may impair grid cell rhythmicity. To establish whether this was the case, we used high-density silicone probes to isolate a total of 279 single-units in layer II/III of the dorsal MEC from 10 mice (WT:150 units from five mice, rTg4510: 129 units from five mice). In this study, we focused on the dorsal MEC, a subregion which has previously been identified as being particularly vulnerable in this transgenic model (*Booth et al., 2016b*). By performing post-hoc electrolytic lesions on each recording shank to identify probe location, we were able to estimate individual unit location by determining the largest average waveform along the 200 μm probe for each isolated unit. Importantly, estimated recording location was not different between genotypes (Median distance from dorsal entorhinal border (interquartile range (IQR)): WT: 350(350) μm, rTg4510: 385(275) μm, U = 9660, p=0.77, Mann-Whitney U test *Figure 2—figure supplement 1*).

Considered across the entire cellular population, MEC single units recorded from rTg4510 mice exhibited a decrease in mean firing rate (Median (IQR): WT: 2.97 (16.7) Hz, rTg4510: 1.03 (3.0) Hz, U = 4472, p=0.01, Mann-Whitney U, n = 150; 129 units, *Figure 2A/B*). As in the CA1 region of the hippocampus (*Booth et al., 2016a*), rTg4510 cells also showed a pronounced decrease in theta modulation of firing patterns (Median (IQR) theta modulation index: WT: 9.17 (13.9), rTg4510: 2.29 (2.81), U = 2296, p<0.001, Mann-Whitney U, n = 150;129 units, *Figure 2C*). Individual units with a theta modulation index (TMI) greater than 5 (*Langston et al., 2010*) were assigned as 'theta

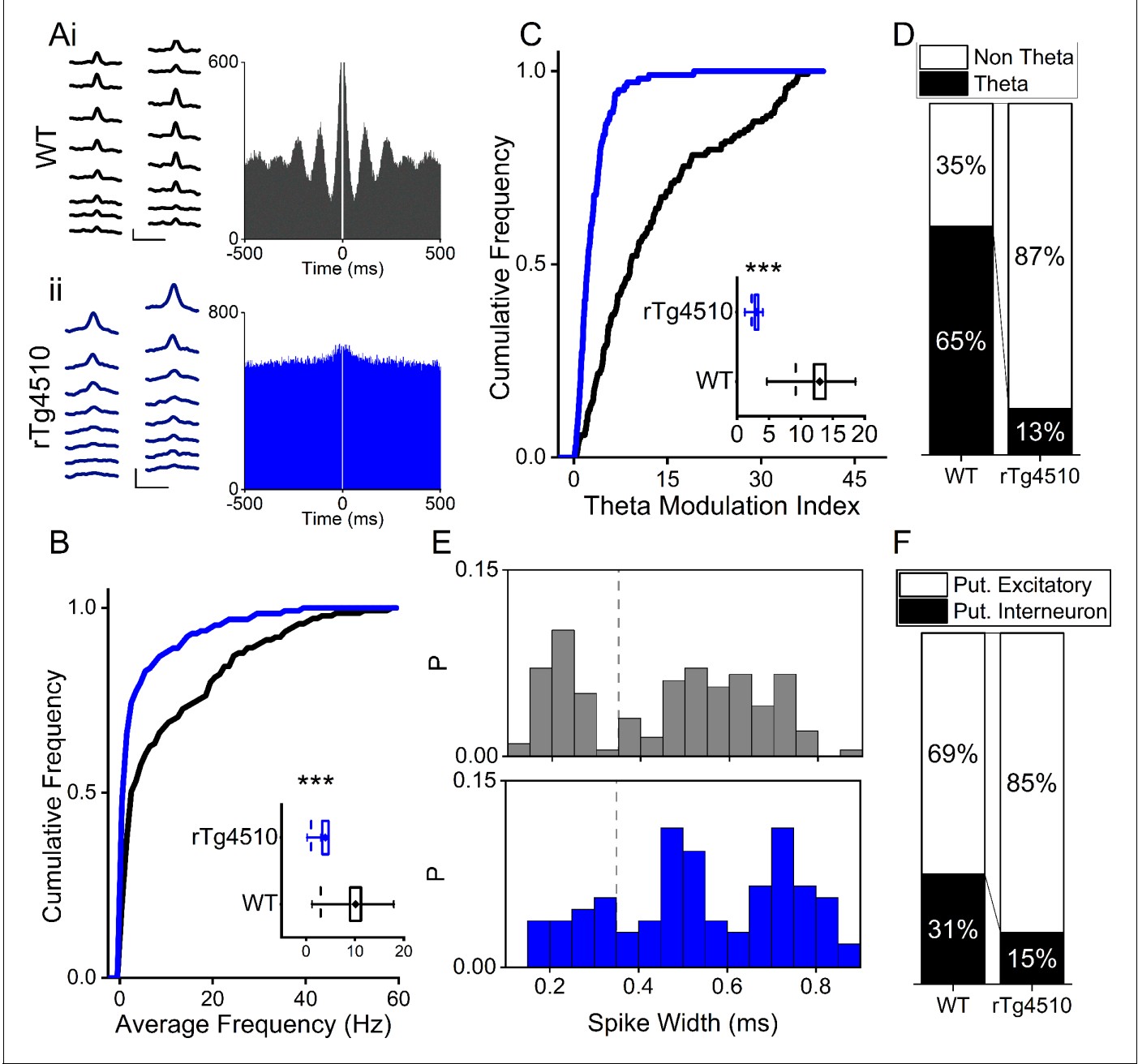

**Figure 2.** Firing properties of mEC single units. (**A**) Average waveforms from an example cell recorded from a 16-channel silicone probe shank for WT (i) and rTg4510 (ii) mouse, with firing autocorrelations. Scale bars: 0.4 ms, 50 µV. (**B**) Cumulative frequency plot of theta modulation index for all recorded mEC single units, with average modulation for WT (black) and rTg4510 (blue) mice inset. (**C**) Proportion of cells displaying theta modulation (threshold: TMI>5). (**D**) Average firing frequency across entire recording session of mEC neurons for WT (black) and rTg4510 (blue) mice, average inset. (**E**) Spike-width probability histogram for WT (black) and rTg4510 (blue) units. (**F**) Proportion of cells classified as putative interneurons (spike-width <0.35 ms, dotted line in (**E**) and putative excitatory. Box plots: dotted line: median, diamond: mean ± SEM, whiskers: 25th/75th centile), *** p <0.001 Mann-Whitney U test.

The online version of this article includes the following figure supplement(s) for figure 2:

**Figure supplement 1.** Recording of mEC single units.

modulated'. Although the majority of WT MEC cells (65%) showed significant theta modulation, only a small proportion (13%) passed threshold in rTg4510 mice ($\chi^2$ (1)=64.1, p<0.0001, Chi-Square test, *Figure 2D*). Extracellularly recorded spike waveforms from interneurons in the mEC have a significantly narrower spike widths when compared to excitatory neurons (*Buetfering et al., 2014*). The probability distributions of single unit spike widths recorded from the dorsal MEC in WT mice had a bimodal distribution (*Figure 2E*). Based on this distribution and previously published work (*Buetfering et al., 2014*), we classed units with spike widths < 0.35 ms as putative interneurons and those >0.35 ms as putative excitatory cells. Using this approach, we found that 47/150 units (31%) were classed as putative interneurons in WT mice, where as in rTg4510 mice a significantly smaller proportion of units were classed as putative interneurons (19/129 units, 15%; $\chi^2$ (1)=16.0, p<0.001, Chi-Square test; *Figure 2E/F*). However, it should be noted that in the presence of tau pathology these measures should be met with caution, since the effect of action potential dynamics under these conditions is not necessarily clear.

## Loss of grid cell periodicity in rTg4510 mice

Consistent with the literature we found that ~1/4 of cells in the WT MEC had grid-like spatial firing patterns in a 0.8 m x 0.8 m square arena (*Figure 3A*). We calculated a grid score for each cell, based on the rotational symmetry of the 2D-autocorrelations and found that 36/150 cells (24%) had grid scores higher than the 95th percentile of the distribution produced from shuffled spike timestamps (threshold = 0.21; *Figure 3B*). In contrast, in rTg4510 mice there was an almost complete breakdown of grid cell periodicity (3/129 units, 2.3%, threshold: 0.26), with animals displaying irregular, non-uniform, firing fields across recording environments ($\chi^2$ (1)=27.1, p<0.0001, Chi-Square test, *Figure 3A/C*). Furthermore, the distribution of grid scores of all cells recorded from WT and rTg4510 mice was significantly different (Median (IQR) grid score: WT, 0.03 (0.3); rTg4510, 0 (0.12), n = 150/129 units; p<0.001; Mann-Whitney U, *Figure 3D*).

We calculated thresholds separately for each genotype because changes to firing rate in particular can have impact on random score distributions. However, using the threshold employed for WT mice (0.21) would result in 6/129 rTg4510 cells (4.7%) classified as grid cells. Furthermore, a combined threshold produced from all the shuffled data (0.23) would have resulted in 4/129 (3.1%) rTg4510 cells classified as grid cells, compared to the three cells identified from the individual thresholds.

Conversely, when measuring the spatial information content (SI) of recorded cells (*Skaggs et al., 1993*), we found that MEC neurons did not differ significantly between genotypes, either as a distribution (WT, 0.36 (0.75) bits/spike, n = 150; rTg4510, 0.49 (0.53) bits/spike, n = 129 units; p=0.38; Mann-Whitney U, *Figure 3E*) or in the proportion of cells that exceed threshold (WT: 12/150, rTg4510 5/129, $\chi^2$ (1)=1.4, p=0.24 Chi-Square test; *Figure 3C*).

## Tau pathology is associated with impaired speed coding in MEC single units

Deficits in running speed-oscillation relationships at a local field potential level suggest alterations in the neural representation of running speed in the MEC of rTg4510 mice. Recent evidence suggests the existence of a separate population of cells in the MEC which express a rate code for running speed ('speed cell') (*Kropff et al., 2015*). We calculated a speed score for all recorded cells, whilst mice ran on an L-shaped linear track, by computing the Fisher-transformed correlation coefficient (z) between instantaneous firing frequency and running speed (*Figure 3A–B*). Units were considered 'speed-modulated' if they had a speed score outside the 5th−95th centile range of a shuffled distribution of data produced from 250 shuffles for each cell (*Figure 3C*). In WT mice, 85/150 (57%) cells had firing rates significantly modulated by running speed, whereas in rTg4510 mice a significantly lower proportion of cells was speed modulated (17/129; 13%; $\chi^2$ (1)=56.55), p<0.0001, Chi-Square test, *Figure 4D*. As an overall population, rTg4510 MEC units also displayed a significantly lower average speed score compared to WT mice (Median (IQR): WT, z = 0.11 (0.31), rTg4510: z = 0.009 (0.01), U = 3603, p<0.0001, Mann-Whitney U, n = 150;129 units, *Figure 4E*), with cells recorded from rTg4510 mice, on average displaying a running speed correlation close to zero.

Speed-modulated MEC neurons can be broadly split into those that display linear and saturating exponential relationships between running speed and firing rate (*Hinman et al., 2016*). To account

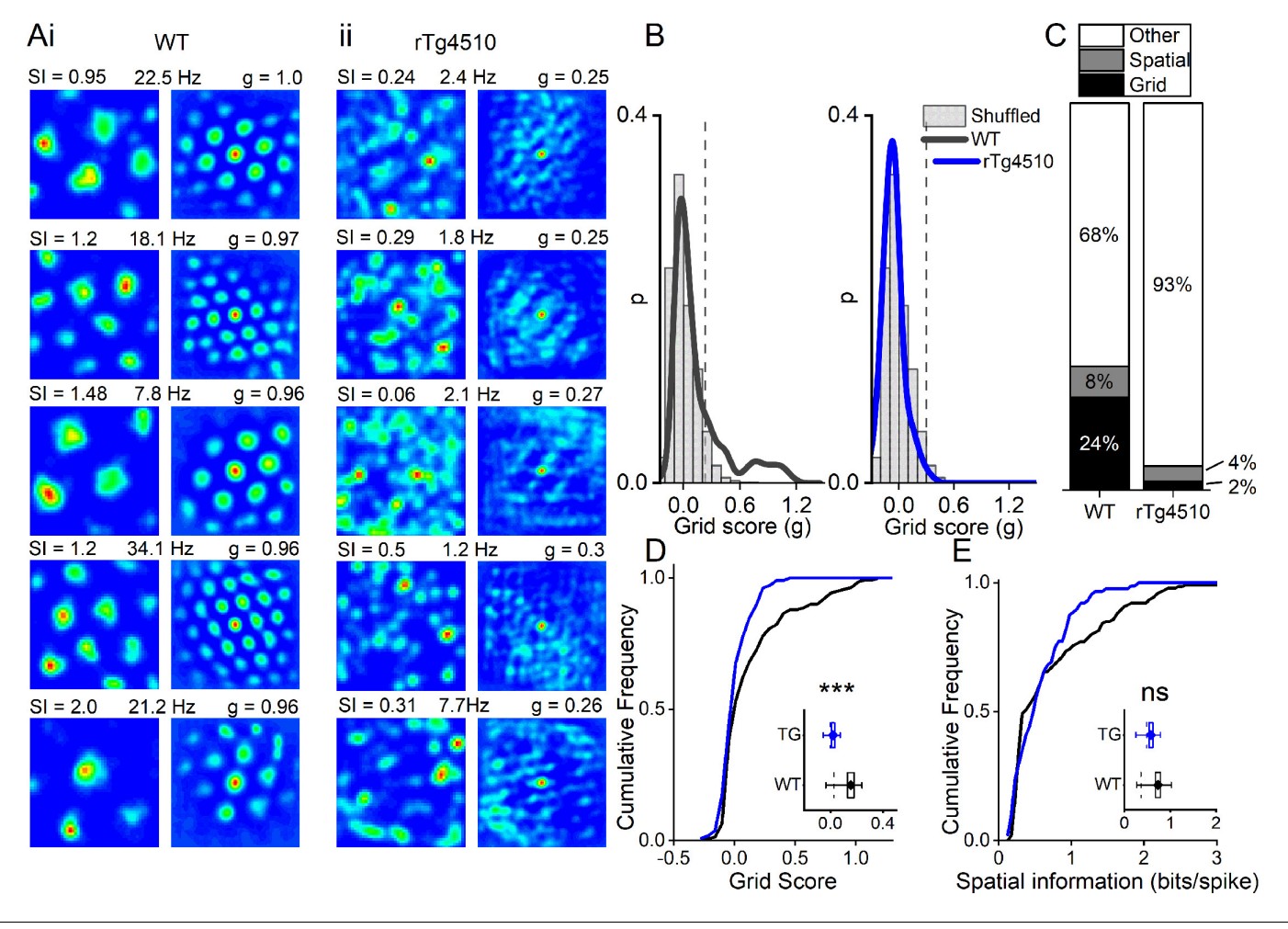

**Figure 3.** Breakdown of grid cell periodicity in rTg4510 mice. (**A**) Example spatial firing patterns (left) and 2D autocorrelations (right) of cells from WT (i) and rTg4510 (ii) mice in a 0.8 x 0.8 m square arena, displayed with grid score (g), spatial information content (SI) and peak firing rate across recording environment. Five cells with the highest grid score are displayed for each genotype, showing irregular firing patterns even in the rTg4510 cells with the highest grid scores. (**B**) Histograms of grid scores for all single units recorded from WT and rTg4510 mice. Observed data plotted as a solid line, shuffled data shown as grey bars. The 95th centile of shuffled distributions is plotted as a dashed line. (**C**) Cumulative frequency plots for grid score; inset box plot illustrating average values for each genotype (dotted line: median, diamond: mean ± SEM, whiskers: 25 th /75 th centile), *** p<0.001, Mann-Whitney U test. (**D**) Proportions of grid and spatial non-grid cells greater than threshold in WT and rTg4510 mice. (**E**) Cumulative frequency plots for spatial information score; inset box plot illustrating average values for each genotype (dotted line: median, diamond: mean ± SEM, whiskers: 25th/ 75th centile), ns p>0.05, Mann-Whitney U test.

for this, speed scores were calculated for log transformed data and further classified as linear or exponential by the best regression fit (*Figure 4—figure supplement 1*). In WT animals, speed-modulated cells showed an approximately equal distribution between linear and saturating running speed relationships (*Figure 4—figure supplement 1* linear: 41/85 units, saturating: 44/85 units), the proportions of which did not differ significantly in the small number of speed-modulated rTg4510 MEC neurons (linear: 10/17 units, saturating: 7/17 units, $\chi^2$ (1)=0.6, p=0.43 Chi-Square test, *Figure 4—figure supplement 1*).

Previous studies have suggested that a small proportion of speed-modulated cells decreased, rather than increased, their firing frequency during locomotor activity (*Kropff et al., 2015*; *Hinman et al., 2016*). On the linear track, in WT mice, this population was observed to be consistent with previous reports (~13% of speed-modulated cells) (*Kropff et al., 2015*; *Hinman et al., 2016*). In contrast, in rTg4510 mice the proportion of cells with negative speed relationships was substantially and significantly higher (WT: 11/85 units, rTg4510: 7/17 units, $\chi^2$ (1)=7.8, p=0.005, Chi-Square test,

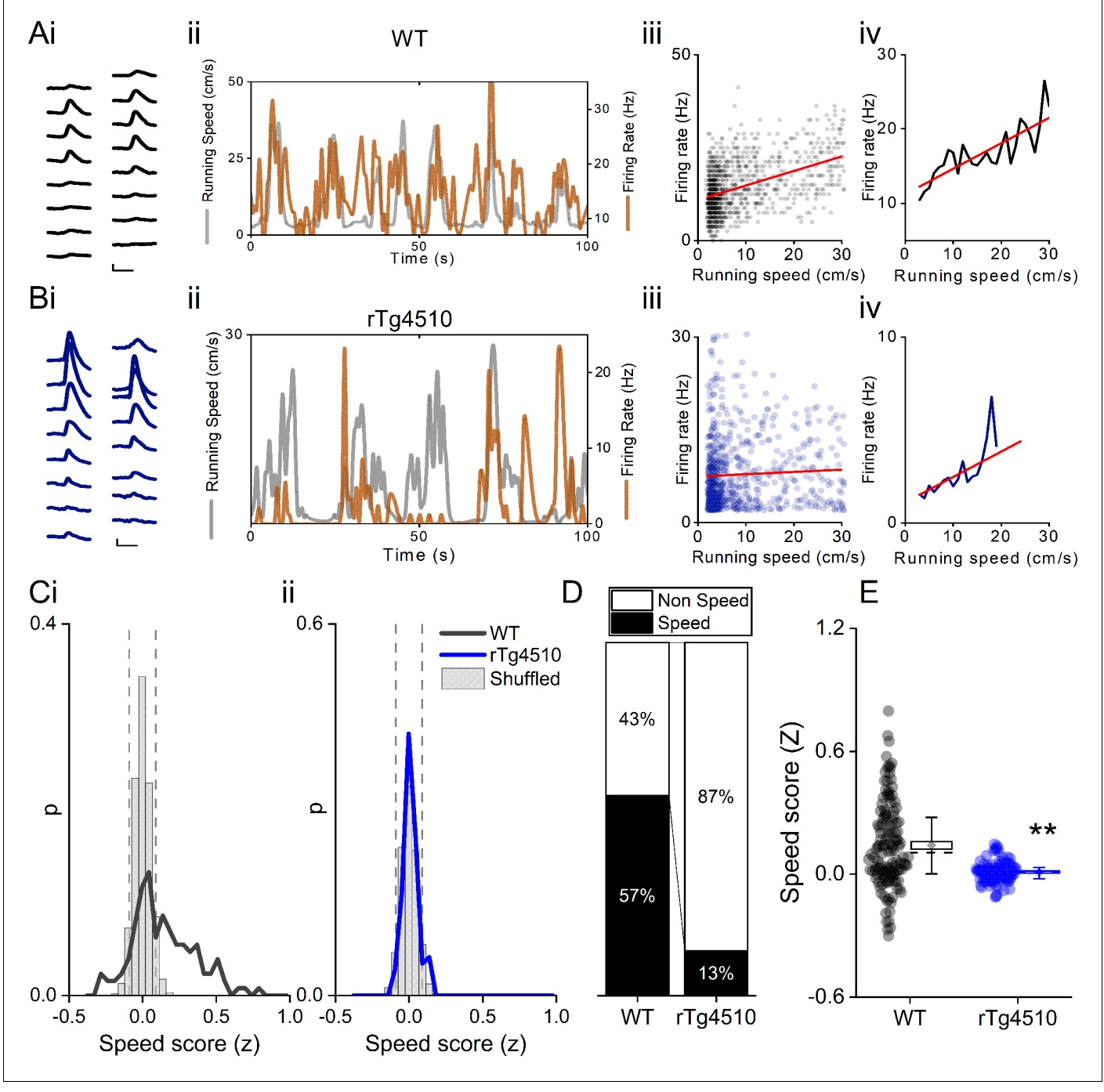

**Figure 4.** Decreased speed modulation of MEC single units in rTg4510 mice. Average waveforms of a single unit recorded from a 16-channel shank silicone probe, from a WT (**A**i; black) and rTg4510 (**B**i; blue) mouse. The running speed (grey) and cell firing rate (orange) (ii), correlation between running speed and firing rate for each time bin (40 ms) (iii), and average for each speed bin (1 cm/s) (iv) are shown for each of these example cells. Red line: linear fit for each. Scale bars: 0.3 ms, 25 µV. (**C**) Distribution of speed scores for WT (i) and rTg4510 units (ii) with shuffled distribution of scores (grey bars); 5th/95th centile threshold: dotted lines. (**D**) Proportion of cells classified as speed modulated (>95th or <5th centile of shuffled distribution). (**E**) Average speed score for each recorded MEC unit. Box plots: dotted line: median, diamond: mean ± SEM, whiskers: 25th/75th centile, **p<0.01, Mann-Whitney U test.

The online version of this article includes the following source data and figure supplement(s) for figure 4:

**Figure supplement 1.** Properties of MEC speed-modulated cells.

**Figure supplement 1—source data 1.** Example images of electrolytic lesions for each mouse.

*Figure 4 continued on next page*

*Figure 4 continued*

**Figure supplement 2.** Speed modulation of single units in hippocampal CA1 pyramidal cell layer also shows similar increase in negatively speed-modulated firing.

*Figure 4—figure supplement 1*), although it should be noted that these proportions are based on small numbers of rTg4510 neurons. In any case, speed-modulated cells were split much more evenly between positive and negative associations with firing rate, as would be expected by chance.

Given that many MEC speed cells are fast spiking in nature (*Kropff et al., 2015*), we also assessed the spiking properties of the recorded of speed-sensitive units. We found that in WT and rTg4510 mice, a similar proportion (41% and 31%, respectively; $\chi^2$ (1)=0.6, p=0.4, Chi-Square test) of speed-modulated neurons were fast-spiking (mean firing rate >10 Hz) (*Figure 4—figure supplement 1*). The proportion of all cells that were classified as fast-spiking was substantially lower in rTg4510 mice (13% vs 30% in WT; $\chi^2$ (1)=19.0, p<0.001, Chi-Square test; *Figure 4—figure supplement 1*), although this likely reflects the overall reduction in mean firing rate in these neurons (*Figure 2E*).

Speed-modulated firing of single units has also been observed in the hippocampus (*Kropff et al., 2015*; *McNaughton et al., 1983*; *Lu and Bilkey, 2009*), so we next sought to determine whether deficits in speed tuning in rTg4510 mice were specific to the MEC, or were also represented downstream in the hippocampus proper. For this purpose, data were taken from previous single unit and local field potential recordings in the hippocampal CA1 region of rTg4510 mice (*Booth et al., 2016a*) at a similar age point and reanalysed to assess the contribution of locomotor activity to firing rate. In this region, CA1 theta band activity in the local field potential was also correlated with running speed in WT mice (linear regression; theta power; WT: $R^2$ = 0.83, p<0.001, n = 6 mice), but not in rTg4510 mice (linear regression: $R^2$ = −0.04, p=0.6, n = 4 mice, *Figure 4—figure supplement 2*). A significant proportion of CA1 neurons had firing rates modulated by running speed in both WT and rTg4510 mice (WT: 25/46 units, rTg4510: 27/52 units). Importantly however, as seen in MEC recordings, a much greater proportion of CA1 cells were negatively modulated by locomotor activity in rTg4510 mice than in WT controls (WT: 5/25 units, rTg4510:13/27 units; $\chi^2$ (1)=4.5, p=0.03, Chi-Square test, *Figure 4—figure supplement 2*).

## Normal head-direction tuning in rTg4510 mice

Another critical component of the path integration system in the MEC are cells which accurately encode directional heading information. Therefore, we next sought to establish whether HD cells (*Giocomo et al., 2014*; *Sargolini et al., 2006*) in the MEC were similarly disrupted in rTg4510 mice. We calculated a HD score by determining the mean vector length of circular firing distributions and, as with other functional metrics, we compared observed HD scores with shuffled distributions (5 *Figure 5A/B*). Importantly, a similar proportion of MEC cells in WT and rTg4510 mice surpassed the 95th-centile threshold (WT: 19/150, rTg4510: 14/129, $\chi^2$ (1)=0.79, p=0.37, Chi-Square test, *Figure 5C*). Furthermore, as a population, the HD score was actually slightly, but significantly, higher in rTg4510 cells compared to WT (Median (IQR) vector length: WT: 0.074 (0.091), rTg4510: 0.11 (0.12), U = 6509, p=0.0014, Mann-Whitney U, n = 150;129 units, *Figure 5B–E*). These data suggest that HD tuning remains intact in rTg4510 mice.

## Decreased firing rate does not account for changes in spatial metrics

Since overall firing rates of MEC neurons in rTg4510 mice are signifcantly slower than WT controls (*Figure 2*), we next asked whether these changes may account for differences in grid, speed, or HD tuning scores between genotypes. Spike trains for each WT cell were downsampled incrementally between 1 and 100 times. As reported previously (*Bonnevie et al., 2013*), healthy grid cells are largely resistent to such decreases in firing, until firing rates are decreased to an average of ~0.1 Hz (*Figure 5—figure supplement 1*). This would represent a 10-fold greater change in firing rate than we observed in rTg4510 MEC neurons (Median (IQR): WT: 2.97 (16.7) Hz, rTg4510: 1.03 (3.0) Hz, U = 4472, p=0.01, Mann-Whitney U, n = 150; 129 units, *Figure 2*).

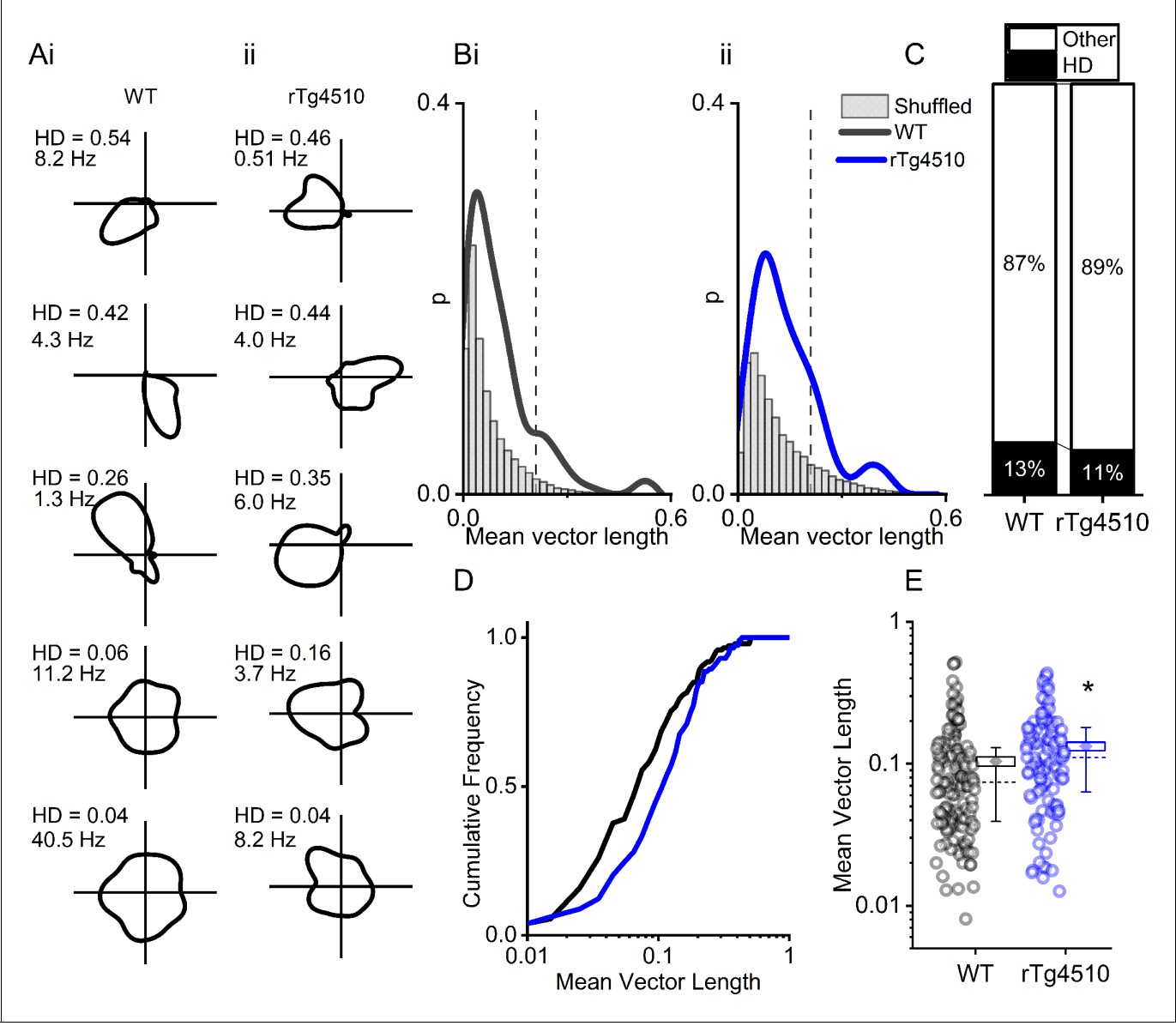

**Figure 5.** rTg4510 mice retain MEC head-direction (HD) tuning. (**A**) Example HD tuning of 5 WT and 5 rTg450 units, displayed with HD score and peak firing rate across HD bins. (**B**) Distribution of HD scores for WT (i) and rTg4510 mice (ii) with shuffled distribution of scores (grey), 95th-centile threshold: dotted line. (**C**) Proportions of cells with HD scores over threshold in WT and rTg4510 mice. (**D**) Cumulative frequency distribution of HD scores (mean vector length). (**E**) Mean vector length for all cells, showing a small but significant increase in HD tuning across the population in rTg410 mice (dotted line: median, diamond: mean ± SEM, whiskers: 25th/75th centile), * p<0.05 Mann-Whitney U test.

The online version of this article includes the following figure supplement(s) for figure 5:

**Figure supplement 1.** Decreased firing rate does not account for changes in spatial metrics.

**Figure supplement 2.** Rate coding stability across recording session.

**Figure supplement 3.** Conjunctive representation of grid, head direction, and running speed in WT and rTg4510 mice.

## Stability of spatial firing in rTg4510 mice

We next tested whether changes in firing properties in rTg4510 mice could be due to a decreased stability of firing across individual sessions. Each session was split into two halves, with scores calculated for each. We found no difference in the average grid score between session halves for either WT (Z = −0.2, p=0.82; Wilcoxon signed ranks test) or rTg4510 mice (Z = 1.6, p=0.11; Wilcoxon signed ranks test). There was also no differences in stability of grid scores between genotypes, with

the difference between the first and second halves of the recording close to zero for both genotypes (Median (IQR) grid score difference (2nd −1 st half), WT: 0.0014 (0.13), rTg4510:−0.011 (0.13), p=0.15, Mann-Whitney U, n = 150; 129 units, *Figure 5—figure supplement 2*). The same was true for speed scores, which were also stable across sessions in both WT (Z = −0.77, p=0.44; Wilcoxon signed ranks test) and rTg4510 mice (Z = 1.6, p=0.10; Wilcoxon signed ranks test) and did not differ between genotypes (Median (IQR) speed score difference (2nd−1st half), WT: 0.005 (0.13), rTg4510: −0.01 (0.1), p=0.54, Mann-Whitney U, n = 150; 129 units, *Figure 5—figure supplement 2*).

However, whilst HD firing appears to be intact in rTg4510 mice, HD scores from transgenic mice were slightly, but significantly increased in the second half of the recording session, compared to the first (Z = −4.35,p=0.0001; Wilcoxon signed ranks test), while WT cells remained constant through each recording session (Z = −1.6, p=0.10; Wilcoxon signed ranks test), indicating that rTg4510 HD cells may take longer to reference to allothetic cues (*Yoder et al., 2011*). Comparing between genotypes, rTg4510 cells were also significantly more variable within sessions when compared to WT controls (Median (IQR) HD score difference (2nd−1st half), WT: 0.01 (0.11), rTg4510: 0.06 (0.14), p=0.001, Mann-Whitney U, n = 150; 129 units, *Figure 5—figure supplement 2*).

The same was true for spatial information scores, with the second half of the session yielding higher spatial information content than the first. Comparing between genotypes, rTg4510 cells were also significantly more variable within sessions when compared to WT controls (Median (IQR) SI score difference (2nd−1st half), WT: 0.036 (0.17) bits/spike, rTg4510: 0.27 (0.63) bits/spike, p=0.001, Mann-Whitney U, n = 150; 129 units, *Figure 5—figure supplement 2*). However, in this case, both genotypes showed increased SI in the second half of the recording session, compared to the first (WT: Z = −6.1,p=0.001; Wilcoxon signed ranks test, rTg4510: Z = −7.1, p>0.0001; Wilcoxon signed ranks test).

## Impaired conjunctive representation of spatial information in rTg4510 mice

A proportion of cells within the MEC are known to demonstrate conjunctive functional representation (*Sargolini et al., 2006*). Using the tuning-curve approaches defined above, we found that ~60% of WT cells passed the threshold for only a single functional class of neuron (i.e. grid-, speed-, or HD-cells). A further ~25% of WT cells passed the threshold for more than one functional class and were therefore conjunctive cells, whilst the remaining ~20% of cells were unclassified (*Figure 5—figure supplement 3*). For example, in this study, only 3% of spatially sensitive cells were pure grid cells, whilst the remaining grid cells also passed the threshold for speed- and HD-modulation (*Figure 5—figure supplement 3*). In contrast, due to the substantial reduction in numbers of speed and grid cells, very few rTg4510 cells (~2%) expressed any conjunctive representation (*Figure 5—figure supplement 3*).

## Discussion

Continuous integration of running speed information in the MEC has been proposed to be critically important for spatial navigation and path integration (*Sargolini et al., 2006*; *Kropff et al., 2015*; *McNaughton et al., 2006*; *Burgess et al., 2007*). The current experiments clearly show that unlike WT animals, rTg4510 mice do not express adequate representations of locomotor activity in the MEC local field potential (LFP), since theta and gamma oscillations display blunted, or absent, relationships with running speed.

There has been some suggestion in the literature that dementia pathology affects the firing pattern of grid cells in the MEC both in mice (*Fu et al., 2017*) and grid-like neural representations in humans (*Kunz et al., 2015*). To date, however, no studies have examined changes to speed-modulated firing of MEC single units in tauopathy. *Fu et al., 2017* also describe no effective impact of tau pathology on HD or border cell firing rates using the rTgTauEC mouse model. However, it should be noted that in this case tau pathology was restricted to the MEC, rather than impacting the broader circuits involved in these processes. Our data suggest the hypothesis that reduced grid cell periodicity may be the result of the impaired integration of running speed information in the MEC. However, this could also originate from a number of other mechanisms, such as a decreased recruitment of fast-spiking interneurons, an impaired spatial coding in the hippocampus or changes to in the intrinsic properties of stellate cells in the MEC. Although the causality of this is not clear, it is evident that

both speed and grid cell signals are perturbed in this tau model, which is likely to have profound influences on spatial information processing.

It is important to note that, as with many other animal models of dementia pathology, we should be cautious about extrapolating phenotypic changes to human pathology. In particular, some of the pathological and behavioural features observed in the rTg4510 have been associated with the specific transgene insertion sites in this mouse line, rather than tauP301L overexpression per se (*Gamache et al., 2019*). In the future, it will be important to explore the impact of tau pathology on spatial coding in alternative, more refined models. For example, a recently described humanised tau mouse line, generated using gene editing approaches, expresses wild-type tau under endogenous promoters (*Saito et al., 2019*). This approach should lead to significantly reduced genetic disruption. Whilst this mouse line does not naturally develop any neuropathological features, it is capable of supporting an accelerated tau pathology when challenged by intracortical injections of human AD brain extract (*Saito et al., 2019*). Exploring hippocampal and cortical spatial coding deficits in such a model would be an important comparator to the current study.

## Vulnerability of grid cell firing

These results highlight the importance of an intact path integration system in maintaining grid cell periodicity. Several studies have shown a breakdown of grid cell firing patterns after the inactivation of important spatial information streams. For example, inhibition of the medial septum, which controls theta rhythmicity, but also speed-modulated inputs, produces a complete breakdown of grid cell periodicity (*Brandon et al., 2011*; *Koenig et al., 2011*). Likewise, inactivation of the anterior thalamic nuclei (ATN) disrupts HD tuning in the MEC, also impairing the grid cell signal (*Winter et al., 2015*). Inactivation of reciprocal hippocampal inputs into the MEC is also sufficient to produce a breakdown in grid cell periodicity (*Bonnevie et al., 2013*).

The almost complete breakdown of grid cell firing in rTg4510 mice contrasts with the effect on spatial representation in the hippocampus of these animals. While several studies have shown a reduction in the spatial information and stability of hippocampal place cells (*Booth et al., 2016a*; *Cheng and Ji, 2013*), firing fields are still consistently present in these mice. Given the pattern of degeneration across the hippocampal formation (*Ramsden et al., 2005*; *Santacruz et al., 2005*; *Spires et al., 2006*; *Blackmore et al., 2017*), poor place representation (*Booth et al., 2016a*; *Cheng and Ji, 2013*) may be the result of weakened entorhinal inputs (*Hales et al., 2014*; *Van Cauter et al., 2008*; *Brun et al., 2008*). Since hippocampal place cells could be thought of as conjunctive integrators of multiple spatial (and non-spatial) input streams, surviving inputs may facilitate encoding of an impoverished spatial representation in these mice (*Booth et al., 2016a*; *Cheng and Ji, 2013*). However, the loss of grid/speed input prevents this from being anchored to an egocentric spatial reference frame, resulting in decreased spatial stability. Presumably, whatever representation is being encoded by place cells is anchored to an allocentric reference frame. Decreasing the availability of visual/contextual cues or implementing a pure path integration task would likely exacerbate spatial learning/memory impairments. It is also possible to consider the role of the lateral entorhinal cortex here, which recent studies have shown to predominately code for egocentric over allocentric information (*Wang et al., 2018*). Reductions in grid and place field activity may therefore mirror their appearance in neuronal development, where place cells appear before grid cells but mature fully only after grid cell development (*Wills et al., 2012*; *Muessig et al., 2015*). The precise temporal nature of grid and place cell impairment in this model is as yet unknown. However, taken together, these data further suggest that hippocampal neurons can form place fields in the absence of effective grid cell firing.

Grid cells are proposed to play a key role in path integration (*McNaughton et al., 2006*; *Burak and Fiete, 2009*; *Fuhs and Touretzky, 2006*; *Etienne and Jeffery, 2004*). For example, mice lacking GluA1-containing AMPA receptors have been shown to display reduced grid cell periodicity, correlating with impairments on a path integration-based task (*Allen et al., 2014*). Path integration has also been suggested to be impaired in populations of dementia and MCI patients (*Allison et al., 2016*; *Hort et al., 2007*; *Mokrisova et al., 2016*). The direct association between these two factors is still unclear; however, it is likely that grid cell deficits, as described above, directly contribute to the deficits in spatial memory in rTg4510 mice (*Booth et al., 2016a*; *Santacruz et al., 2005*).

While we are confident that we recorded the activity of MEC single units, it is also possible that variations in electrode placements across genotypes could have influenced our results. Incorrectly

placed electrodes alone could account for a reduction in grid cells in the rTg4510 group. The pronounced cortical shrinkage produced by this model makes the precise positioning of electrode arrays difficult to achieve using standard stereotaxic coordinates.

## Firing rate changes in rTg4510 mice

These data show large differences in firing rates between WT and rTg4510 mice, which may have the capacity to bias measurements such as the grid score. However, while most manipulations that disrupt grid firing also decrease entorhinal firing rates, it is important to note that 'healthy' grid scores are largely resistant to moderate to large decreases in firing rate. By down sampling the spike trains of MEC neurons we show that decreased firing alone is not sufficient to account for changes in spatial metrics seen here. It is more likely, for example, that changes to theta rhythmicity, which often accompany the breakdown of grid cell periodicity, are responsible for changes in these mice.

## Stability of head-direction tuning

A further feature of the inactivation studies described above (*Bonnevie et al., 2013*; *Brandon et al., 2011*; *Koenig et al., 2011*; *Winter et al., 2015*) is the stability of HD tuning in the MEC. Blocking medial septum activity impairs grid cell firing, but not HD tuning (*Brandon et al., 2011*; *Koenig et al., 2011*). Interestingly, the breakdown of grid periodicity after hippocampal inactivation has also been shown to reveal HD tuning in grid cells that would previously not have been modulated by HD (*Bonnevie et al., 2013*). It is possible that the small but significant population increase in HD tuning in rTg4510 mice may reflect this unmasking of HD inputs from cells that would have previously displayed grid patterns. However, this is a hypothesis that would need to be addressed directly with long-term recordings of grid cells as tau pathology incrementally increased to levels suffient to impair spatial firing patterns. This approach is unlikely to be achieved with current electrophysiological technology. In any case, the arrival of HD information into the MEC is likely the result of an anatomically distinct pathway (*Taube, 2007*) that appears to be relatively preserved in the face of the tau pathology burden in rTg4510 mice at this stage. This is most likely due to a greater dependence on subcortical structures, such as the ATN, that integrate vestibular information (*Stackman and Taube, 1998*; *Blair et al., 2007*; *Sharp et al., 2001*).

Interestingly, while directional tuning, in general, appears to remain intact after tau deposition of this kind, the stability of HD scores was significantly decreased in rTg4510 mice. In this case, scores were on average higher during the second half of recording sessions, suggesting a subtle temporal deficit in rate coding. This may be such that, in rTg4510 mice, HD sensitivity takes a period of time to become fully evident, consistent with disorientation upon entering what should be a familiar environment.

## Impaired running speed representations in rTg4510 mice

Theta oscillations exhibit increases in both amplitude and frequency depending on running speed (*Figure 1*). Spatially modulated MEC neurons are required to integrate large quantities of multimodal sensory information from their environment. At fast running speeds, the time window for this integration is smaller and it may therefore be necessary to increase the sensitivity of such neurons during locomotion to accurately retrieve spatial associations from memory. In rTg4510 mice, theta oscillation power appears to be independent of running speed meaning that these animals are unable to integrate this information effectively.

In WT mice, the gamma oscillation power relationship with running speed shows a strong predominance for fast gamma frequencies (*Figure 1*). This is consistent with the proposed information flow across the hippocampal formation, in which the MEC provides the input responsible for fast gamma frequencies in CA1 region of the hippocampus (*Colgin et al., 2009*). While the CA1 area has been shown to display two distinct peaks in the gamma band power spectra, MEC LFP predominantly contains faster gamma frequencies only (*Colgin et al., 2009*; *Chrobak and Buzsáki, 1998*). It is perhaps not surprising then, that differences between genotypes (*Figure 1*) are only observed at these faster frequencies. For this reason also, conclusions regarding the slow gamma oscillation frequency should be made with caution. While slow gamma frequency has previously be shown to display a negative speed relationship in the hippocampus, MEC correlations are generally thought to be positive (*Zheng et al., 2015*; *Kemere et al., 2013*).

It is also worth noting that rTg4510 mice, like several other models of dementia, exhibit a hyperactive phenotype (*Blackmore et al., 2017*; *Jul et al., 2015*; *Cheng et al., 2014*; *Przybyla et al., 2016*), and hence altered patterns of locomotor activity. This may reflect dysregulation of the septo-hippocampal pathway, activation of which has been shown to stimulate movement or alternatively may reflect dysfunction in motor-related brain regions. Nevertheless, whilst it should be noted that rTg4510 mice are experiencing their environment at a generally higher state of arousal, at least in terms of locomotor activity, the key finding of this study is that this locomotor activity is improperly represented in the hippocampal formation.

## Conclusions

Overall, these data show a clear breakdown in grid cell periodicity in rTg4510 mice compared to WT controls. In addition, they suggest a role for the dysfunctional processing of locomotor activity in this process, since the representation of running speed information in MEC single units is severely disrupted in these mice, while HD tuning remains constant, or slightly increased. The changes to MEC single unit firing are likely to have profound implications for the impairments in spatial memory observed in these mice and suggest observable parameters to assess in dementia patient populations, for example through speed-modulated fMRI signals in virtual environments.

# Materials and methods

**Key resources table**

| Reagent type (species) or resource | Designation | Source or reference | Identifiers | Additional information |
|---|---|---|---|---|
| Strain, strain background (*M. musculus,* Male) | rTg4510, Tg(Camk2a-tTA)1Mmay Fgf14 | ENVIGO | RRID:MGI:4819951 | Gift from Eli Lilly. (*Ramsden et al., 2005*) |
| Software, algorithm | Klusta suite | *Rossant et al., 2016* | RRID:SCR_014480 | https://klusta.readthedocs.io/ |
| Software, algorithm | MATLAB R2019b | Mathworks https://uk.mathworks.com/ | RRID:SCR_001622 Chronux toolbox: RRID:SCR_005547 | |
| Software, algorithm | OriginPro 2019b | OriginLab https://www.originlab.com/ | RRID:SCR_014212 | |
| Other | Cresyl violet stain | Sigma aldrich | ID: C5042 | |
| Other | High-density silicon probe electrode array | Cambridge NeuroTech https://www.cambridgeneurotech.com/ | P2 | two × 16 channel shanks |
| Other | Silicon Probe | Neuronexus https://neuronexus.com/ | A1 × 16–5 mm-150-703 | 16 channel linear array |
| Other | Digital Lynx 10S recording system | Neuralynx https://neuralynx.com/ | HS-18 or HS-36 | Cheetah five data acquisition software |

## Animals

All procedures were carried out in accordance with the UK Animal (Scientific Procedures) Act 1986 and were approved by the Universities of Exeter and Bristol Animal Welfare and Ethical Review Body (PPL P29FAC36A).

The rTg(tet-o-TauP301L)4510 mouse model (*Ramsden et al., 2005*; *Santacruz et al., 2005*) was bred on a mixed FVB/NCrl + 129S6/SvEvTa background and delivered to the University of Exeter via Envigo (Loughborough, UK). Male rTg4510 and age-matched littermate WT mice were housed on a 12 hr light/dark cycle with ad libitum access to food and water. rTg4510 mice express a repressible form of human tau containing the P301L mutation that has been linked with familial frontotemporal

dementia. They represent one of the most well characterised models of tauopathy (*Santacruz et al., 2005*; *de Calignon et al., 2010*; *Spires-Jones et al., 2011*).

## Surgical implantation

All surgical procedures were conducted using standard sterile and aseptic techniques. Animals were anaesthetised using isoflurane (4%) and fixed into a stereotaxic frame (ASI instruments). Anaesthesia was reduced and maintained at 1–2% during surgery. After careful cleaning of the skull surface, small screws (Antrin Miniature Specialities) were inserted into each bone plate in order to anchor the electrode array. Silver wire (World Precision Instruments) was soldered to a screw overlying the cerebellum to be used as a ground.

Probes were implanted at 0.2–0.3 mm anterior to the transverse sinus and 3–3.25 mm from midline. Linear probes were implanted and fixed 3 mm below the dura mater and angled at 10 degrees in the posterior to anterior direction in the sagittal plane in order to record consistently from layer II/III along the dorsal-ventral axis of the MEC. High density 16- (Neuronexus) or dual shank 32-channel (Cambridge Neurotech) silicone probes were implanted 0.3–0.5 mm below dura at an angle of 5°, also in the anterior to posterior direction and subsequently moved slowly into the cortex using their attached microdrive (Cambridge NeuroTech). RelyX Unicem two dental cement with blue curing light (Henry Schein) were used to anchor the probe to the skull and anchor screws.

## Data acquisition

Animals were given at least 1 week of post-operative recovery before initial recording sessions. Local field potential (LFP) signals were recorded using a Digital Lynx 10S recording system (Neuralynx, Bozeman, MT, USA) tethered to a HS-18 or HS-36 unity gain headstage and Cheetah five data acquisition software (Neuralynx). The headstage and tether were counterbalanced using a moveable, weighted arm to allow for the maximum flexibility of movement. Two light-emitting diodes (LEDs) on the headstage and an overhead video camera (sample rate 25 Hz) were used to continuously track the animals' location using Cheetah's built in video tracking software (VTS), allowing estimation of position and therefore running speed. Once recorded, invalid tracking points, that is time-points where no light threshold was reached, were excluded and the animal's position interpolated from the two nearest points. Estimation of running speed was performed on binned position data, with erroneous bins, above 50 cm/s, also removed. LFP data were recorded while animals explored either a linear track (1.5 m) or square open field (0.8 m x 0.8 m).

For single-unit experiments, high-density arrays were moved slowly through cortex until reliable units were seen across all channels corresponding with activity in layer II/III MEC. In some animals, after recording sessions, electrodes were moved further into the cortex and data obtained from a second recording location. In these case locations were separated by at least 200 μm from the bottom of the first location to the top of the second, as determined by fractions of a micro screw (i.e. by moving the electrode array at least 400 μm in total) in order to be confident that that same neurons where not being sampled from.

## Analysis of local field potential (LFP) signal

Data were continuously sampled at 2 kHz, band-pass filtered (1–500 Hz) and stored on a PC for offline analysis. All LFP signals were analysed in MATLAB, using open-source toolboxes or custom routines utilising built-in functions. Multi-tapered spectral analysis was performed using the Chronux toolbox (available at http://chronux.org/). The most dorsal MEC channels (as determined by post-hoc histology) were used for LFP analysis.

Power and peak frequency of LFP frequency bands were compared to running speed. Spectral analysis was conducted on 0.5 s bins of LFP data and compared to running speed calculated from the same time frame. For running speed curves, locomotor activity was divided into 1 cm/s bins (between 1 and 30 cm/s) and oscillatory power and peak frequency averaged across all relevant sections of data. Theta and gamma oscillation power was normalised to the power in these frequency bands during non-movement, defined as speeds under 1 cm/s.

## Analysis of single-unit data

For single unit data, recordings were referenced to the ground electrode, continuously sampled at 40 kHz, bandpass filtered between 1 and 30 kHz and saved unprocessed on a PC for offline analysis. Each channel was referenced offline to a common-average of the opposite 16-channel shank (250 μm away) in order to eliminate signals common across the electrode array such as noise and movement artefacts. Extracellular spike activity was detected and sorted using the klusta open source software package found at: http://klusta.readthedocs.io/en/latest/ (*Rossant et al., 2016*). Cluster isolation was determined by the cluster quality metric from the KlustaViewa software (*Rossant et al., 2016*), cluster were included if the score was greater than 0.95.

Clusters were classified as either putative interneurons or putative excitatory cells (pyramidal or stellate cells) by their spike half-width, taken from the peak to the subsequent trough of the average extracellular waveform. While the majority of cells recorded in the MEC are excitatory, a significant population can be classified as inhibitory interneurons (*Buetfering et al., 2014*; *Beed et al., 2013*; *Miettinen et al., 1996*). Using the average spike waveform, putative interneurons were classified as displaying a spike-width less than 0.35 ms, based on the extracellular properties of PV+ interneurons isolated optogenetically (*Buetfering et al., 2014*). This approach was taken alone, rather than in combination with average firing rare of neurons, since MEC interneurons have been shown to vary widely in their spike frequency (*Buetfering et al., 2014*). Neurons were described by a theta modulation index (TMI), based on the fast Fourier transform (FFT) of spike-train autocorrelations, using methods described previously (*Booth et al., 2016a*; *Langston et al., 2010*; *Wills et al., 2010*). Autocorrelations were produced with ± 500 ms lags and 2 ms bin size. The peak at 0 lag was reduced to the next maximal value and the entire function mean-normalised by subtracting the mean from all values. The autocorrelation was tapered using a Hamming window to reduce spectral leakage and FFT calculated. The power spectrum was calculated by dividing the square of the FFT by the transform length ($2^{16}$, scaled to the length of the autocorrelation). TMI was defined as the mean power within 1 Hz of each side of the peak in the theta frequency range (5–12 Hz) dived by the mean power between 0 and 125 Hz. Cells were defined as 'theta modulated' if their TMI was greater than 5.

## Analysis of speed modulated firing

Speed modulation of single unit activity was calculated based on analysis described in *Kropff et al., 2015*. Running speed and firing rate of individual clusters were calculated for 40 ms bins of data and smoothed across 500 ms using a Gaussian window function. Running speeds from 2 to 30 cm/s and containing more than 0.5% of total recording duration were used for further analysis. Speed modulation of cells was then defined by the correlation between all running speed and firing rate bins and a speed score calculated using the Fisher-z transformation of the correlation coefficient, r. Observed speed correlations were compared to a distribution of randomly sampled correlations of shuffled data. For shuffling, time stamps were forward-shifted by a pseudorandom period between 20 s and the total trial length minus 20 s, with the end of the trial wrapped to the beginning and reanalysed using the method above. Cells were defined as 'speed modulated' if their speed score (z) was greater than the 95th percentile, or less than the 5th percentile, of the global distribution of scores produced from at least 250 shuffled data sets for each unit (*Figure 3*).

## Analysis of head direction properties

HD was determined by calculating the angle between two LEDs attached to the animal's headstage. Time periods where neither, or only one, of the LEDs were observed above threshold were discarded. Firing rate was calculated for 3° bins of HD and smoothed, using a Gaussian window over 14°. A 'HD score' was defined as the resultant mean vector length, calculated from the smoothed firing rate histograms. Observed mean vector length was also compared to the 95[th] percentile of a distribution of shuffled data produced as above.

## Analysis of spatial firing properties

Spike locations for each cell were obtained with a 2D histogram count, using the MATLAB function *histcounts2*. Firing rate was calculated for 3 × 3 cm bins across recording environments and smoothed using a 2D Gaussian function across 1.5 standard deviations. 'Gridness' was calculated using a 2D autocorrelation of smoothed firing rate maps (*Sargolini et al., 2006*). Spatial periodicity

was determined by rotating autocorrelations in steps of 30°, between the central peak and the six closest peaks, and correlating the rotated versions with the original. Grid score was expressed as the difference between rotations at 30°, 90°, and 150° with the central peak removed, where if firing maps show a hexagonal pattern give low correlations, and 60° and 120° where correlations will be high.

The spatial information content (SI) of each cell was defined using the measure described by *Skaggs et al., 1993* and expressed in terms of bits/spike. This approach measures the extent to which a cell's firing rate can be used to predict the animal's location. By definition, this does not assume spatial periodicity and has been used for quantifying place cell activity (*Booth et al., 2016a*; *Cacucci et al., 2008*; *Brun et al., 2008*; *Hussaini et al., 2011*) as well as spatially selective firing in the lateral EC (*Deshmukh and Knierim, 2011*).

Cells were defined as grid cells if their grid score was greater than the 95th percentile of the global distribution of scores produced from at least 250 shuffled data sets for each unit. Cells were defined as spatially modulated if their SI score exceeded the 95th percentile of a distribution of shuffled data but did not pass the criteria for grid cell classification.

## Downsampled firing properties

To control for changes in neuronal firing rate effecting coding properties, spike trains for each WT cell were uniformly downsampled incrementally between 1 and 100 times. Grid scores, mean vector lengths, and speed scores were then calculated again for each case and compared to rTg4510 scores at equivalent firing rates.

## Hippocampal CA1 data

Single unit and LFP data recorded from the CA1 pyramidal cell layer from previously published datasets were additionally analysed for speed encoding (*Booth et al., 2016a*). These data were collected using microdrives containing independently adjustable tetrodes, while animals ran on a linear track. Running speed–firing rate relationships were determined using the processing pathway described above.

## Electrode placement

At the end of experiments, mice received a lethal overdose of sodium pentobarbital (Euthetal) and electrolytic lesions were made at several electrode locations across the recording array. Mice were then transcardially perfused with 4% v/v formaldehyde in 0.1 M phosphate buffered saline (PBS). Brains were extracted from the skull and stored in 4% formaldehyde before being cut in sagittal sections (50 μm) using a vibratome (Leica VT1000) and stained with cresyl violet. The position of electrode sites was determined from digital pictures taken with a 2.5X objective on a light microscope using QCapture pro seven software (Qimaging). Probe electrode location was expressed as distance from the most dorsal electrode site in layer II/III MEC.

## Acknowledgements

This work was supported by an Alzheimer's Research UK Major Project Grant awarded to JB (ARUK-PG2017B-7). JW was an Alzheimer's Research UK Fellow (ARUK-RF2015-6). AR was a Royal Society Industry Fellow. KP is an employee of Eli Lilly and Company. Eli Lilly provided all the mice used in this study.

## Additional information

### Competing interests
Keith G Phillips: is an employee of Eli Lilly. The other authors declare that no competing interests exist.

## Funding

| Funder | Grant reference number | Author |
|---|---|---|
| Alzheimer's Research UK | ARUK-PG2017B-7 | Thomas Ridler<br>Andrew D Randall<br>Jonathan T Brown |
| Eli Lilly and Company | PhD Studentship | Keith G Phillips<br>Andrew D Randall<br>Jonathan T Brown |
| Alzheimer's Research UK | ARUK-RF2015-6 | Jonathan Witton |
| Royal Society | Industry Fellow | Andrew D Randall |

The funders had no role in study design, data collection and interpretation, or the decision to submit the work for publication.

### Author contributions

Thomas Ridler, Conceptualization, Formal analysis, Investigation, Visualization, Methodology, Writing - original draft, Writing - review and editing; Jonathan Witton, Investigation, Writing - review and editing; Keith G Phillips, Resources, Supervision, Methodology, Writing - review and editing; Andrew D Randall, Resources, Supervision, Funding acquisition, Writing - review and editing; Jonathan T Brown, Conceptualization, Resources, Data curation, Formal analysis, Supervision, Funding acquisition, Writing - original draft, Project administration, Writing - review and editing

### Author ORCIDs

Thomas Ridler (iD) https://orcid.org/0000-0002-8236-9033
Andrew D Randall (iD) https://orcid.org/0000-0001-8852-3671
Jonathan T Brown (iD) https://orcid.org/0000-0001-5269-7661

### Ethics

Animal experimentation: All procedures were carried out in accordance with the UK Animal (Scientific Procedures) Act 1986 and were approved by the Universities of Exeter and Bristol Animal Welfare and Ethical Review Body. PPL P29FAC36A.

### Decision letter and Author response

Decision letter https://doi.org/10.7554/eLife.59045.sa1
Author response https://doi.org/10.7554/eLife.59045.sa2

## Additional files

### Supplementary files

• Transparent reporting form

### Data availability

All data will be made available via The Center for Open Science (https://osf.io/83yfd/).

The following dataset was generated:

| Author(s) | Year | Dataset title | Dataset URL | Database and Identifier |
|---|---|---|---|---|
| Ridler T | 2020 | Impaired speed encoding is associated with reduced grid cell periodicity in a mouse model of tauopathy | https://osf.io/83yfd/ | Open Science Framework, 83yfd |

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
