## [Decision Letter]

**Acceptance summary:**

The manuscript convincingly shows that the neuronal representation of running speed and the periodic firing of grid cells are impaired in a mouse model of tauopathy. These results will likely be of interest to those studying spatial coding in the medial entorhinal cortex and also those who are studying navigational deficits in neurodegenerative disorders.

**Decision letter after peer review:**

Thank you for submitting your article "Impaired speed encoding is associated with reduced grid cell periodicity in a mouse model of tauopathy" for consideration by *eLife*. Your article has been reviewed by three peer reviewers, including Kevin Allen as the Reviewing Editor and Reviewer #1, and the evaluation has been overseen by Laura Colgin as the Senior Editor. The following individuals involved in the review of your submission have agreed to reveal their identity: James Alexander Ainge (Reviewer #2); Martin Fuhrmann (Reviewer #3).

The reviewers have discussed the reviews with one another and the Reviewing Editor has drafted this decision to help you prepare a revised submission.

We would like to draw your attention to changes in our revision policy that we have made in response to COVID-19 (https://elifesciences.org/articles/57162). Specifically, we are asking editors to accept without delay manuscripts, like yours, that they judge can stand as *eLife* papers without additional data, even if they feel that they would make the manuscript stronger. Thus the revisions requested below mainly address clarity and presentation. Additional analyses were also requested when judged essential.

Summary:

The manuscript investigates how spatial coding is affected in the medial entorhinal cortex of a mouse model of tauopathy. High-density silicone probes and tetrodes were used in freely moving mice to characterize the activity of speed cells, grid cells, and head-direction cells. Encoding of running speed, both in the local field potentials and in the spiking activity of neurons, was significantly impaired in rTg4510 mice. Grid cell periodicity was also altered but head-direction coding appeared largely intact. This study provides a clear demonstration that speed coding is impaired in a mouse model of tauopathy and, therefore, points to potential new mechanisms for understanding navigation deficits in Alzheimer's patients.

In general, the reviewers were positive about the quality of the experiments and the novelty of the findings. Concerns were raised regarding some methodological details that were missing in the manuscript. Additional issues were raised concerning the interpretation of some of the findings, mainly related to the small population of head-direction cells with negative speed-rate correlations.

We encourage the authors to implement the essential revisions and to send us their revised manuscript.

Essential revisions:

1) The demonstration that grid periodicity is lost in rTg4510 mice is based on recordings taken in only five mice per group. To judge the significance of this finding, it is critical to know how reliably grid cells could be recorded in control and rTg4510 mice. The authors should provide a table showing how many cells of the different functional classes (grid cells, head-direction cells, etc.) were recorded in each mouse. Ideally, in this type of experiment, one would use mice as statistical units when comparing groups. The same applies to speed scores. The reader should know how many cells are recorded from each animal and whether the effects were driven by specific animals or they generalized across all animals.

2) The number of speed cells detected in the MEC of rTg4510 is very close to chance lever (10% vs. 13%), and the mean speed score is close to 0. The distribution of speed scores in rTg4510 mice is very similar to the distribution obtained by chance (Figure 3Cii). The analysis in subsection “Tau pathology is associated with impaired speed coding in MEC single units” is presented as though there are more negatively correlated speed cells in rTg4510 mice than in control: "speed modulated cells were split much more evenly between positive and negative associations with firing rate." However, given that the number of speed cells in rTg4510 is very close to chance levels, the more even split might not be due to similar positive and negative speed coding in rTg4510 mice, but rather to the absence of significant speed coding.

3) Towards the end of the Results section, the authors present a number of analyses based on very small numbers of cells. For example, examining the proportion of the 17 speed modulated cells in the TG mice that are negatively modulated. Or examining border cells in the two groups. Unless the authors can make a convincing case for the robustness of these findings, we would recommend either taking these out or making it clear that these findings should be interpreted with caution.

4) One limitation that should be discussed related to the use of the rTg4510 mouse line. A recent study by Gamache et al., (2019) has shown that overexpression of tauP301L in these mice is not sufficient to cause the premature and robust age-related loss of forebrain mass and gross forebrain atrophy. Indeed, mice with similar expression levels of tauP301L but different tauP301L insertion sites in the genome, atrophy, and tau histopathology are delayed, and a different behavioral profile is observed. It is, therefore, possible that the phenotype observed in the current manuscript is not only due to the expression of tauP301L but also to the dysregulation of other genes. While comparable problems are likely to affect other Tau mouse models, we think this limitation should be stated somewhere in the discussion. This could highlight the importance of replicating the current findings in a different model of tauopathy.

5) The title of the manuscript suggests that the impairment in grid cell periodicity is somehow linked to the impaired speed coding. While both are present in rTg4510, the evidence for a link between these two phenomena is rather weak. For example, there is no correlation between these two variables in this study. I think the manuscript would be stronger with less emphasis on the potential mechanistic link between impaired speed coding and grid cell periodicity before the Discussion section. For example, the title could be more neutral regarding the association between grid periodicity and speed coding. Perhaps "Impaired speed encoding and grid periodicity in a mouse model of tauopathy." The sentence “We propose a model whereby such deficiencies in speed encoding networks result in loss of grid cell firing patterns in the MEC” in the Introduction could also be modified. In rTg4510 mice, the impairment in grid cell periodicity could originate from several mechanisms, including a decreased recruitment of high-firing rate interneurons, an impaired spatial coding in the hippocampus, a reduction in speed coding, a change in the intrinsic properties of stellate cells in layer II, etc. The demonstration of an impaired speed coding in rTg4510 mice is very interesting, even without a clear causal link to grid cells.

6) From which layers of MEC were recordings made – this is not clear from Figure 2—figure supplement 1A. Ideally, the field of view would be more extensive. In the last picture in rTg4510 mice, the electrode lesions appear to be more medial, perhaps in the parasubiculum.

7) There are a number of details of the experiments missing. How many sessions of recording were the data collected from? Are all the data from a single session or are the data collapsed across multiple sessions? If the data are from a single session how is this session chosen? If it is from multiple sessions is there an effect of experience – i.e. is the deficit in speed coding stable across sessions or does this change?

8) Are speed/grid and HD cell recordings taken from multiple days? If so, how do the authors control for the possibility of recording from the same cells on multiple occasions which would artificially increase the sample size?

9) Is cluster quality measured? If not, how do the authors control for effects being driven by poorer cluster isolation in the rTg4510 mice?

10) The shuffling procedure for grid cell classification is standard but in the current study this results in a higher threshold for grid cells for the rTg4510 mice than the WT mice. Does this skew the results?

11) How are the examples in Figure 2 chosen? Are these the best ones? Are they representative of the data set? Are they from different mice? Examples should be representative of the main findings rather than being cherry-picked.

12) Please include the distribution of spatial information scores in Figure 2. The spatial information scores in the examples in Figure 2 are much higher for the wild types than the rTg4510. It would be good to see what the cells with high spatial scores in the rTg4510 mice look like.

13) The lack of difference in spatial information scores reported in the text and Figure 2D is puzzling. The figure shows that 92% of rTg4510 cells are non-spatial compared to only 69% of the WT cells. How do the authors reconcile these apparently contradictory findings?

14) Related to the above point, it is not stated what the threshold for classifying a cell as spatial is.

15) There are a number of reasons why grid scores may be lower in the rTg4510 mice. One of these is that grids might drift over time. If this was the case the grid pattern may be present if shorter sessions were used. This could be examined by analyzing gridness in the first and second half of the sessions independently.

16) Related to the above point – how consistent is spatial firing across sessions? Were multiple sessions recorded? If so, are the spatial patterns consistent? If not, this could be assessed by correlating the first and second half of each session.

17) The dataset (spike times, position data, LFP data) should be made freely available together with the code used for the analysis (unless there are good reasons to restrict access).

---

## [Author Response]

Essential revisions:1) The demonstration that grid periodicity is lost in rTg4510 mice is based on recordings taken in only five mice per group. To judge the significance of this finding, it is critical to know how reliably grid cells could be recorded in control and rTg4510 mice. The authors should provide a table showing how many cells of the different functional classes (grid cells, head-direction cells, etc.) were recorded in each mouse. Ideally, in this type of experiment, one would use mice as statistical units when comparing groups. The same applies to speed scores. The reader should know how many cells are recorded from each animal and whether the effects were driven by specific animals or they generalized across all animals.

As suggested, we have added this information in the form of a table added to Figure 2—figure supplement 1.

2) The number of speed cells detected in the MEC of rTg4510 is very close to chance lever (10% vs. 13%), and the mean speed score is close to 0. The distribution of speed scores in rTg4510 mice is very similar to the distribution obtained by chance (Figure 3Cii). The analysis in subsection “Tau pathology is associated with impaired speed coding in MEC single units” is presented as though there are more negatively correlated speed cells in rTg4510 mice than in control: "speed modulated cells were split much more evenly between positive and negative associations with firing rate." However, given that the number of speed cells in rTg4510 is very close to chance levels, the more even split might not be due to similar positive and negative speed coding in rTg4510 mice, but rather to the absence of significant speed coding.

This is an excellent point. We have removed the section in the Discussion speculating on negative speed modulation in rTg4510 mice. Since negative speed cells are a feature of locomotor encoding in multiple brain region we have kept the analysis of these cells in the results but have highlighted the small numbers in the text and that this outcome would be expected by chance (Results).

3) Towards the end of the Results section, the authors present a number of analyses based on very small numbers of cells. For example, examining the proportion of the 17 speed modulated cells in the TG mice that are negatively modulated. Or examining border cells in the two groups. Unless the authors can make a convincing case for the robustness of these findings, we would recommend either taking these out or making it clear that these findings should be interpreted with caution.

We agree that for the reasons stated above it is somewhat speculative to include the analysis of border cells, due to the low number of detected cells in this functional group. Therefore, we have removed the section on border cells but have kept the negative speed cell analysis (see above), if only to highlight that those that pass the threshold do so close to chance levels for positive and negative distributions.

4) One limitation that should be discussed related to the use of the rTg4510 mouse line. A recent study by Gamache et al., (2019) has shown that overexpression of tauP301L in these mice is not sufficient to cause the premature and robust age-related loss of forebrain mass and gross forebrain atrophy. Indeed, mice with similar expression levels of tauP301L but different tauP301L insertion sites in the genome, atrophy, and tau histopathology are delayed, and a different behavioral profile is observed. It is, therefore, possible that the phenotype observed in the current manuscript is not only due to the expression of tauP301L but also to the dysregulation of other genes. While comparable problems are likely to affect other Tau mouse models, we think this limitation should be stated somewhere in the discussion. This could highlight the importance of replicating the current findings in a different model of tauopathy.

This is a good point and agree it is worthwhile highlighting the potential problems associated with the use of transgenic mice as models human disease. The Gamache paper makes this point in rather a stark fashion, so we have added a paragraph discussing these limitations to the Discussion. In this paragraph we also highlight a possible alternative model which may be a useful comparator group.

5) The title of the manuscript suggests that the impairment in grid cell periodicity is somehow linked to the impaired speed coding. While both are present in rTg4510, the evidence for a link between these two phenomena is rather weak. For example, there is no correlation between these two variables in this study. I think the manuscript would be stronger with less emphasis on the potential mechanistic link between impaired speed coding and grid cell periodicity before the Discussion section. For example, the title could be more neutral regarding the association between grid periodicity and speed coding. Perhaps "Impaired speed encoding and grid periodicity in a mouse model of tauopathy." The sentence “We propose a model whereby such deficiencies in speed encoding networks result in loss of grid cell firing patterns in the MEC” in the Introduction could also be modified. In rTg4510 mice, the impairment in grid cell periodicity could originate from several mechanisms, including a decreased recruitment of high-firing rate interneurons, an impaired spatial coding in the hippocampus, a reduction in speed coding, a change in the intrinsic properties of stellate cells in layer II, etc. The demonstration of an impaired speed coding in rTg4510 mice is very interesting, even without a clear causal link to grid cells.

As suggested, we have changed the title to remove the causal implication between the impairments in speed and grid coding in the MEC. Furthermore, we have removed the sentence in the Introduction and added it to the Discussion.

6) From which layers of MEC were recordings made – this is not clear from Figure 2—figure supplement 1A. Ideally, the field of view would be more extensive. In the last picture in rTg4510 mice, the electrode lesions appear to be more medial, perhaps in the parasubiculum.

We apologise for this omission. Recordings were made from layer II/III and we have now added this information to subsection “Analysis of single unit data”. We were not able to be more specific than this as some of the ex vivo lesions were quite large making it difficult to define layers in some cases, nevertheless, we are confident that all of our recording sites are located in the MEC. In a number of animals the electrodes were also moved after recording in order to find more cells – so lesions are in final location rather than recorded.

7) There are a number of details of the experiments missing. How many sessions of recording were the data collected from? Are all the data from a single session or are the data collapsed across multiple sessions? If the data are from a single session how is this session chosen? If it is from multiple sessions is there an effect of experience – i.e. is the deficit in speed coding stable across sessions or does this change?

The data from each neuron analyzed came from a single session. We used a procedure where we continued to move through the cortex between recordings in order to maximize our ability to sample from the most optimal MEC layers. Where more than one recording location was used, sessions were picked to ensure all electrodes were a suitable distance from each other (>200 µm) so we were confident that the same cells were not recorded multiple times. Added to the Materials and methods to clarify this.

8) Are speed/grid and HD cell recordings taken from multiple days? If so, how do the authors control for the possibility of recording from the same cells on multiple occasions which would artificially increase the sample size?

Some animals have more than one recording location, which are separated by >200µm from the bottom of the first location to the top of the second. Added to Materials and methods. We have also added this information to the table in Figure 2—figure supplement 1.

9) Is cluster quality measured? If not, how do the authors control for effects being driven by poorer cluster isolation in the rTg4510 mice?

We used the cluster quality metric from the KlustaViewa software (Rossant et al., 2016) with a threshold of 0.95, which did not alter between groups. Clarified in subsection “Analysis of single unit data”.

10) The shuffling procedure for grid cell classification is standard but in the current study this results in a higher threshold for grid cells for the rTg4510 mice than the WT mice. Does this skew the results?

We calculated thresholds separately for each genotype since changes to firing rate in particular can have impact on random score distributions. However, using the threshold for wildtype mice would result in 6/129 cells classified as grid cells. Furthermore, a combined threshold produced from all of the shuffled data would have resulted in 4/129 compared to the 3 cells identified from the individual thresholds. Added to subsection “Loss of grid rhythmicity periodicity in rTg4510 mice”.

11) How are the examples in Figure 2 chosen? Are these the best ones? Are they representative of the data set? Are they from different mice? Examples should be representative of the main findings rather than being cherry-picked.

We deliberately chose cells with the highest grid scores from each genotype to display. This was done in order to highlight the lack of periodicity in the rTg4510 cells since even those with high enough scores to pass threshold do not have visible grid patterns. This was in the legend but we have elaborated on this slightly to clarify our reasoning for selecting these cells. The cells displayed are from 3 mice for each genotype.

12) Please include the distribution of spatial information scores in Figure 2. The spatial information scores in the examples in Figure 2 are much higher for the wild types than the rTg4510. It would be good to see what the cells with high spatial scores in the rTg4510 mice look like.

Cumulative frequency plot for spatial information score has been added to Figure 3E, showing the distribution of spatial information

13) The lack of difference in spatial information scores reported in the text and Figure 2D is puzzling. The figure shows that 92% of rTg4510 cells are non-spatial compared to only 69% of the WT cells. How do the authors reconcile these apparently contradictory findings?

We believe this arises from the amount of spatial information (SI) in a grid cell and the sensitivity of the grid score. Some grid cells (with large spacing) will have high SI scores, while those with smaller spacing will be conveying less spatial information and therefore will have lower scores. Therefore, not all grid cells with pass the threshold for SI and vice versa.

It is also possible for more Tg4510 cells to convey small amounts of SI. If each grid cell were to breakdown, it would almost immediately fail to pass the grid score threshold (which is very sensitive to hexagonal firing patterns). However, the cell may still include a reasonable amount of spatial information.

14) Related to the above point, it is not stated what the threshold for classifying a cell as spatial is.

Thank you for pointing out this omission. We have now added this to the Materials and methods. We used the same “tuning curve” approach as for other cell types i.e. ninety fifth centile of shuffled distribution but also not a grid cell.

15) There are a number of reasons why grid scores may be lower in the rTg4510 mice. One of these is that grids might drift over time. If this was the case the grid pattern may be present if shorter sessions were used. This could be examined by analyzing gridness in the first and second half of the sessions independently.

Thank you for this useful suggestion. We have carried out this analysis as suggested and present the data in Figure 5—figure supplement 2 and in the text (Discussion). Our analysis suggests that for grid- and speed-cells there was no significant change in the rate coding properties between the first and second half of the sessions in either the wildtype or the rTg4510 mice. Interestingly when we carried out the same analysis in for head direction scores, we found that, whilst in WT mice there was no change in HD score between the two halves of the session, in the rTg4510 mice there was an overall increase in HD score. So despite the fact that head direction tuning overall appears to be intact, these analysis uncover a subtle temporal deficit, such that in rTg4510 mice the head-direction sensitivity takes a few minutes to become evident.

16) Related to the above point – how consistent is spatial firing across sessions? Were multiple sessions recorded? If so, are the spatial patterns consistent? If not, this could be assessed by correlating the first and second half of each session.

Our analyses on individual units came single recording session so it is not possible to carry out these analyses suggested here. However, based on the above result, in the future it would be interesting to examine stability of head direction tuning across sessions in the rTg4510 mice.

17) The dataset (spike times, position data, LFP data) should be made freely available together with the code used for the analysis (unless there are good reasons to restrict access).

Processed and source data will be available on the Center for Open Science framework repository at: https://osf.io/83yfd/.